# Profiling of Antibiotic Residues in Surface Water of River Yamuna Stretch Passing through Delhi, India

Suriyah Akhter [1], Mohd Aadil Bhat [2],*, Sirajuddin Ahmed [3],*, Weqar Ahmad Siddiqi [1], Sayeed Ahmad [4] and Hitesh Shrimal [5]

1   Department of Applied Science and Humanities, Jamia Millia Islamia, New Delhi 110025, India
2   School of Ocean and Earth Science, Tongji University, 1239 Siping Road, Shanghai 200092, China
3   Department of Environmental Science and Engineering, Jamia Millia Islamia, New Delhi 110025, India
4   Department of Pharmacognosy and Phytochemistry, Jamia Hamdard, New Delhi 110062, India
5   Waters India Pvt. Limited, New Delhi 110025, India
*   Correspondence: aadilgeo@tongji.edu.cn (M.A.B.); suahmed@jmi.ac.in (S.A)

**Abstract:** The River Yamuna is one of the largest rivers in northern India. It serves as a major source of potable water to the National Capital Delhi. A study was carried out to monitor and quantify three common antibiotics in surface water, with the help of Ultra-Performance Liquid Chromatography (UPLC) coupled with the Mass spectrophotometer (MS/MS)-based method. All the major river water locations in the research area were sampled in triplicate to determine the significant presence of antibiotics from local areas, hospitals, drug manufacturing facilities, and residential areas. Using spiked, as well as actual samples, an optimized method for the detection and quantification of different concentrations of antibiotics was developed. Spatiotemporal variations in the physicochemical properties at the five sites, including reference site-6 (Jamia Millia Islamia), during the pre-monsoon and post-monsoon period, were also investigated. Maximum concentration was observed during the pre-monsoon season followed by the post-monsoon season. The highest detected antibiotic concentration from the river water samples was that of ofloxacin (145.3794 ng/mL), followed by amoxicillin (3.033 ng/mL) and erythromycin (2.171 ng/mL). An aquatic environment may be very vulnerable to dangers from these pharmaceutical residues, according to this risk assessment. Additionally, the levels and compositions of pharmaceutical residue in the aquatic ecosystem might be influenced by seasonal agricultural application, precipitation, and temperature. The findings revealed the existence of antibiotics because of their indiscriminate usage, which may have resulted in the development of resistant strains and ultimately contributed to the spread of antibiotic resistance among living organisms in the river. Thus, attention is required particularly to this section of river as it is the local lifeline source for urban consumers for the domestic water supply and farmers for cultivation.

**Keywords:** surface water; antibiotics contamination; Yamuna river; liquid chromatography-mass spectrophotometry; risk analysis

## 1. Introduction

In an aquatic environment, the issue of antibiotic residues is emerging as a cause for concern because of their presence in significant quantities, unchecked consumption, and improper disposal, all of which are responsible for their extensive presence [1]. There is very little information about the harmful effects of antibiotics and their residues on humans, animals, and other living organisms [2]. All this is because of their widespread and indiscriminate usage in veterinary, medicinal, and agricultural procedures and for the treatment of different diseases [3]. Even though continuous exposure to low dosages of drugs can lead to the development of antibiotic resistance, antibiotics are still a serious cause of concern because of their persistent nature, incomplete metabolism, and accessibility of translocation, via different environmental compartments [4]. About 50–90% of

human and animal metabolism is partial. This has resulted in the widespread detection of waste water and receiving surface water bodies as antibiotics are excreted in parent and metabolite form through feces and urine [5]. The use of antibiotics has significantly increased life expectancy and lowered death rates. However, as the full extent of antibiotics' negative environmental impacts is still unknown, widespread antibiotic use has led to major environmental problems [6]. Water and soil are the major environmental matrices where antibiotic residues have been detected often in recent years, along with sediments and other biological samples [7]. The Yamuna River in Delhi is reportedly constantly polluted for a variety of reasons, including the poor quality of sewer collectors, a lack of wastewater treatment facilities, and untreated sewage from slums and illicit companies [8–10]. Use of antibiotics as therapeutic agents, as well as growth promoters, in the livestock industry, is very common nowadays. Excreted antibiotics found in animal manure are frequently spread on agricultural fields, where these may pollute the groundwater through seeping and finally reach surface water bodies such as rivers and lakes. Their increased penetration into drinking water could have led to undesirable health impacts on the entire population [11]. For instance, it has been observed that the drug carbamazepine induces an oxidative effect on the cells from non-target species, such as mussels, and impairs their health. It has been revealed that some antibiotics cause pathogenic resistance, making the treatment of diseases extremely difficult [12]. As wastewater treatment plants are not made to entirely remove pharmaceuticals, it is essential to keep a check on the treatment of infectious diseases caused by antibiotics and their metabolites that are present in recycled surface water. This can be achieved by reducing their concentration to a varying degree [13]. Most of the surveys conducted by China after 2005 showed that antibiotic contamination in natural surface waters was at the level of ng/L and, in sewage treatment plants, hospital wastewaters, and aquaculture waters, these are at a level of ng/L–μg/L [14]. In India, treated sewage was found to contain an extremely high percentage of antibiotics [15]. Most of the global research conducted on the antibiotic contamination in aquatic environments has focused on rivers, sewage treatment plants (STPs), aquaculture waters, hospital wastewaters, coastal waters, and groundwater while studying antibiotic contamination in aquatic environments [16]. However, the fact is that surface water and wastewater have also been found to contain antibiotics and their metabolites and there are still no systematic solutions that will prevent these water bodies from such pollution [17]. From country to country, the amount and consumption pattern of antibiotics vary. Thus, the possible risks associated with antibiotic residues could be different in India [18]. Antibiotics, such as amoxicillin, erythromycin, ofloxacin, and ampicillin, are largely manufactured and, thus, increase the chance of release in the aquatic environment and are also most commonly reported in the surface and wastewater samples [3]. Historically, several guidelines for environment risk assessment were proposed by different agencies. Hence, risk assessments should also be considered.

## 2. Global Literature Survey of Antibiotics from Surface Waters

Among the most significant antibiotics entering in the aquatic ecosystem are those from surface runoff. The first instance of antibiotic contamination of surface water was reported in 1980 in England, where at least one of the antibiotics from each group of macrolide, sulfonamide, and tetracycline was discovered in quantities of 1000 ng/L in river water [19]. From the study of surface waters of Madrid, Macao, and Guangzhou, various antibiotics such as erythromycin, clarithromycin, ciprofloxacin, norfloxacin, enrofloxacin, and lomefloxacin were identified [20]. Currently, one of the world's major concerns has been the lack of access to clean water as most of the surface water supplies are laced with antibiotics. Fluoroquinolones, tetracycline, sulfonamides, peptidyl transferase, and nitroimidazole were among the antibiotics detected in drinking water and were found to have a value of 0.05–21.4 ng/L, and river water had a concentration of between 0.03 and 3.9 ng/L [21]. A high concentration of antibiotics in river water was also shown by studies conducted, especially in China and India, where no standard prescription is required for their sale [22].

Advanced analytical technologies have enabled the detection of antibiotics at extremely low concentrations (ng/L–µg/L) in different water matrices such as surface waters, wastewater effluents, soil, sediments, and sludge [21,22]. For example, Carbamazepine is found in all the three ecosystems of aquatic environments (wastewater, surface water, and groundwater). Because of its resistance to biodegradation, it is not removed completely by treatment plants [23]. There is a need to identify high-priority pharmaceuticals having the potential to cause a major impact on the environment, as the pharmaceutical residues are ubiquitously present and have potential implications in the aquatic environment affecting the water quality and lower organisms. As they are reportedly present in lower concentrations, their effect cannot be seen immediately but are accumulating, which may have chronic effects on living creatures in the near future [21,22]. Table S2 presents the literature survey conducted on the surface water ecosystems in many parts of the world. It can be seen that antibiotics were the most commonly found pollutants in these water bodies.

The targeted liquid chromatography mass spectrophotometry (LC-MS) method has been widely used to quantify the varying concentration of antibiotics in surface water ecosystems [24]. For the analysis of antibiotics and their metabolites in different environmental samples, highly sensitive equipment is required. In recent years, liquid chromatography, in combination with mass spectrophotometry (LC-MS), has been the most preferred choice of researchers. Although gas chromatography is known to have great resolution power, liquid chromatography (LC) is often favored, as most antibiotics are volatile, heat-sensitive, and contain polar chemicals that cannot be identified using the GC process without a derivatization step, although are known to have a high-resolution power. This is to ensure that antibiotics can be analyzed using LC, without the complicated derivatization process [25].

Thus, determining the presence and amount of the targeted antibiotics in the chosen Yamuna River sites was the primary goal of the current study, combined with the creation of a quick and reliable technique for several antibiotics and to evaluate the ecological risks based on the risk quotient. The general details of the identified pharmaceutical substances and their CAS numbers are provided in Table S1.

## 3. Materials and Methods

### 3.1. Study Area

The Yamuna River originates from the Yamunotri Glacier and flows for about 1376 km, draining a region of about 366,220 km$^2$. The Yamuna river, being the second largest tributary of the river Ganga, provides practical support for irrigation, industry, and urbanization, in addition to supporting a wide variety of species [9,26,27]. Most of the drinking water along its route comes from it. The Yamuna River, which runs next to Delhi, India, for 22 km, was the site of the current study. Through the Wazirabad barrage, the Yamuna reaches Delhi. It then travels through the Okhla barrage to join the Agra canal (Figure 1). According to reports of the CPCB [8], the Yamuna river's water quality is good all the way upstream to Wazirabad in Delhi from its source. Once it enters Delhi, it becomes one of the most polluted river length stretches from Wazirabad to Okhla, which makes up less than 2% of the river's entire length, and it receives the majority of its pollution load from sewage and industrial discharges [28]. Photosynthesis cannot take place in this part of the Yamuna River, due to the low water quality. Instead, the settling of organic materials is the main method used to reduce the biological oxygen demand (BOD) [29]. At many sites, the Yamuna River exhibits seasonal fluctuations in its various physicochemical properties. According to studies, the Yamuna River in Delhi's upper reaches is of higher quality than that in its lower levels. In accordance with established APHA guidelines, a number of physicochemical properties were also assessed in order to learn more about the Yamuna river's water quality [9,10,26,30]. The investigation was performed to find out where some of the regularly used antibiotics ended up in the Delhi section of the Yamuna River. These antibiotics were chosen because these have been found in wastewater samples. These antibiotics are produced in significant quantities due to the strong demand and local

consumption, which also increases the likelihood of these being released into the aquatic environment [3]. Guidelines for the environmental risk assessment (ERA) of pharmaceuticals were introduced by the U.S. Food and Drug Administration and reported in [31,32]. The Environmental Risk Assessment of Pharmaceuticals for Human Use state that the EMA Guidelines on evaluation of potential environmental concerns should be carried out in the following order: Pre-screening, Screening, and Extended [9,33]. The danger is indicated by a coefficient (log $K_{ow} > 4.5$), which denotes the possibility of drug chemicals entering organisms from the aquatic environment. Technical Guidance Documents in Support of the Commission Directive 93/67/EEC on Risk Assessment for New Notified Substances and the Commission Regulation (EC) 1488/94 on Risk Assessment for Existing Substances should be used to screen it for potential for persistence, bioaccumulation, and toxicity. In this study, ecological risk assessments were carried out by determining the risk quotient of selected drugs and also in accordance with the previous studies [9,34]. Between Wazirabad and Okhla catchment areas of the River Yamuna, five sampling locations, designated as Site-1 to Site-5, were selected and one reference sampling location, Site-6 (Jamia Millia Islamia), was also chosen to monitor the varying concentration of the selected antibiotics.

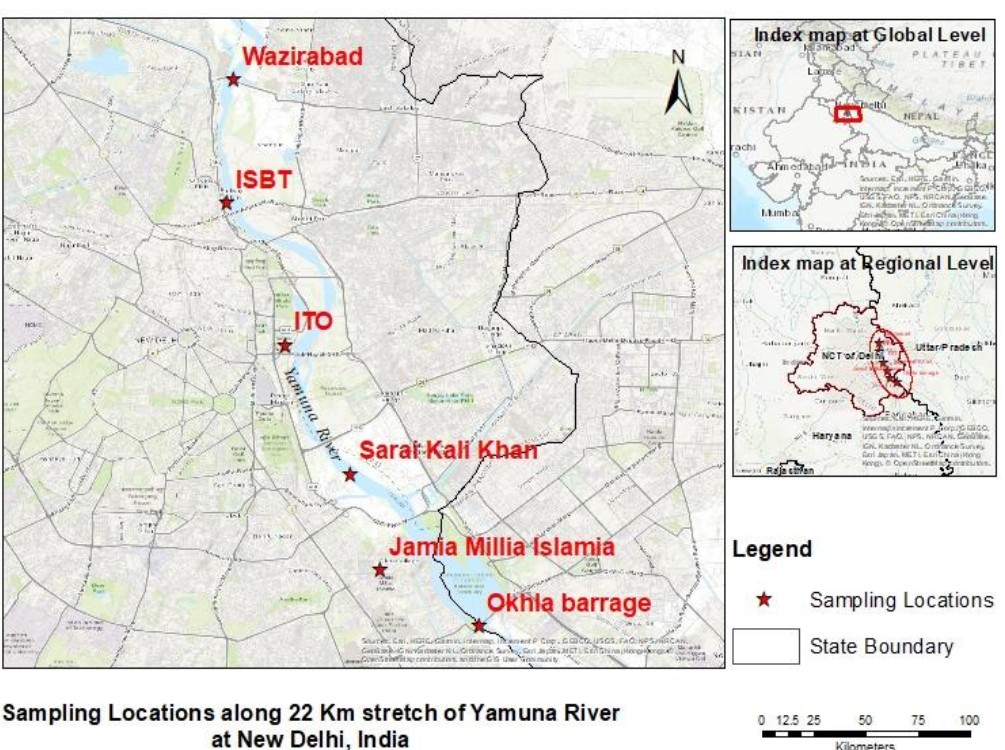

**Figure 1.** Sampling locations of surface water of the River Yamuna.

### 3.2. Sampling Location

Five distinct sampling locations were selected along a 22 km stretch of the Yamuna River that runs through Delhi from Wazirabad to Okhla. From the surface, water samples were taken. Upstream of the Wazirabad barrage was the first sample location (Site-1). This is from where Delhi's drinking water is generated. Sampling at this site provided information on the water's quality prior to the arrival of wastewater from catchment areas. The remaining four spots were chosen to represent the effects of wastewater discharge from various sources. Site-2, the second site for sampling, was the Inter State Bus Terminal (ISBT). The third sample point was ITO (Site-3), the fourth was Sarai Kale Khan (Site-4), wherefrom the Okhla STP treated sewage is also released, and the final sampling point was that of Okhla barrage (Site-5), which is downstream of the third sampling point. Additionally, one water sample from Jamia Millia Islamic was taken, as a reference (Site-6). The description of sampling locations is given in Table 1, along with their geographical coordinates. The

sampling was carried out in accordance with the established protocol [35]. Proper safety measures were taken to prevent any organic material loss while sampling, handling, or transport. Pre-monsoon (April–May) and post-monsoon (November-December) seasons where the two time periods in which sampling was performed. An amount of 5 L of virgin-quality amber (dark)-colored HDPE plastic bottles were used to collect the samples. The obtained sample bottles were delivered to the lab within 24 h of collection and kept in dark, airtight, ice-cold containers.

**Table 1.** Description of the Yamuna River sampling sites.

| Sampling I.D | Sampling Area | Site-Description | Latitude | Longitude |
|---|---|---|---|---|
| **Site-1** | Wazirabad barrage | Initial sampling point | 28.7116 | 77.235 |
| **Site-2** | ISBT | 9.2 km from wazirabad barrage | 28.67417 | 77.23235 |
| **Site-3** | ITO | 14.9 km from wazirabad barrage | 28.6308 | 77.2506 |
| **Site-4** | Sarai Kali Khan | 18.6 km from wazirabad barrage | 28.5915 | 77.27115 |
| **Site-5** | Okhla barrage | 21.5 km from wazirabad barrage | 28.54538 | 77.31149 |
| **Site-6** | Jamia Millia Islamia | Reference sampling point | 28.5623 | 77.2804 |

### 3.3. Chemicals, Standards, and Reagents

In this study, the high-purity standards of antibiotic compounds, selected for analysis, were purchased from Sigma-Aldrich chemicals, Bengaluru (India). The highest purity grade was used for each standard that was employed in this study. Apart from water, all the solvents used for liquid–liquid extraction (LLE) and UPLC-MS analysis were of UPLC quality. The calibration standards and reagents were prepared using Milli-Q water. Filtration was conducted using disposable syringes with a 3 mL capacity and 0.45 µm pore-size sterilized syringe filters as described by [1]. To extract the antibiotics, analytical-grade chloroform was used. The laboratory's distillation equipment provided the distilled water needed for the rotary evaporator.

### 3.4. Ultra-Performance Liquid Chromatography (UPLC) Conditions

Using a Waters UPLC H-Class separation module, the liquid chromatogram separation was carried out (Milford, MA, USA). Likewise, the ACQUITY UPLC BEH C18 (2.1 mm i.d. × 50 mm, particle size 1.7 m) was used. LC columns were evaluated using 0.1 percent formic acid in ultrapure water as eluent "A," whereas 0.1% formic acid in ultra-pure acetonitrile was used as eluent "B". The produced mobile phase was ultrasonically processed through 0.22 m sterilized cellulose nitrate filter paper after being sonicated at room temperature for 10 min. To improve the approach, numerous runs were made under various circumstances. These included variable flowrates, temperatures, and mobile phase compositions. The gradient that produced the best separation began with 90 percent of eluent A and was held for 1 min to 90 percent of eluate A. From 1 to 2 min, it was gradually changed from 90 to 50 percent of eluate A, and for another 2 min to 20 percent of eluate A, which was held for another one minute and then changed to 90 percent of Eluent A in 0.1 min, which was then kept at the same condition at 90 percent eluent A for another 2 min. The injection had a 10 µL volume. A Waters UPLC H-Class with a quaternary pump having a Photodiode detector (PAD; ACQUITY UPLC BEH C18 (2.1 mm i.d. × 50 mm, particle size 1.7 m)) was used. The LC column and auto sampler for auto injection were used to perform the tandem mass analysis. With the use of these instruments, the procedure was optimized and a universal method for identifying all three chosen antibiotics was created. Seven distinct concentrations, ranging from 5 ng/mL to 200 ng/mL, were made for each standard stock solution. Each concentration was passed through the UPLC at least thrice, to check the stability, peak areas, and the retention time of the system. Plotting of the concentration versus peak area produced calibration curves. Through stock solutions with larger concentrations of pure antibiotic standards, the specificity was achieved. The absence of contaminants was confirmed by the single sharp peak, which also assisted in calculating the retention time. This was performed five times to confirm the peak area

and the retention time. Analyzing actual and spiked (selected antibiotics) water samples helped to further confirm the procedure. The standard deviation of response and slope of the curve were used to assess the limit of detection (LOD) and limit of quantitation (LOQ) using the calibration curve, i.e., LOD = 1.5 ng/mL; LOQ = 5 ng/mL [36,37].

Mass Spectrophotometer

Using liquid chromatography and the Waters Xevo TQ-S Micro and UPLC H-Class, the medication concentration was analyzed (Tandem Quadrupole Mass Spectrometer). A positive-mode electrospray ionization source (ESI) assists the UPLC in identifying and quantifying the chemical. To optimize the dependent parameters, a direct infusion of 10 μL/min of standard solution of all the analytes was employed. The next step was the selection of ionization and precursor ion modes. The product ions and their corresponding capillary voltage was 4.5 kV, the source temperature was 150 °C, the desolvation temperature was (450 °C), the cone gas flow was (50 L/h), and the desolvation gas flow was (850 L/h). All these were determined using the automated quantitative optimization method [38,39]. Simultaneously, the analytes quantification and confirmation transitions were observed, and the precursor ion and the product ions selected for the quantification transition showed the strongest response and highest sensitivity. Table 2 presents a brief list of the ideal mass transitions obtained from the UPLC-MS analysis.

**Table 2.** Mass transitions (UPLC-MS) of antibiotics in river water of the Yamuna.

| Antibiotics | Calibration Curve | $R^2$ | Precursor Ion (*m/z*) | Product Ion (*m/z*) | LOD (ng/mL) | LOQ (ng/mL) |
|---|---|---|---|---|---|---|
| **Ofloxacin** | y = 269.05x + 1172.69 | 0.992 | 362.3 | 318.3 | 1.5 ng/mL | 5 ng/mL |
| **Amoxicillin** | y = 221.43x + 227.59 | 0.999 | 366.2 | 114 | 1.5 ng/mL | 5 ng/mL |
| **Erythromycin** | y = 644.30x + 673.29 | 0.997 | 734.5 | 158.1 | 1.5 ng/mL | 5 ng/mL |

The following were the ideal run circumstances for the sample analysis: With a precise flow rate of 0.35 mL/min, 10 μL of the sample was injected into the UPLC column. As the setting for the column was "not controlled," the temperature was 40 °C. Concentrations of antibiotics were measured by comparing the peak areas of actual samples with the peaks of standard solutions, as described in Equation (1), and the response factor was estimated by using Equation (2) [1,40].

$$\text{Analyte amount} = \frac{\text{Peak Area of the Sample}}{\text{Response Factor}} \tag{1}$$

$$\text{Sensitivity factor} = \frac{\text{Peak Area of the Standard}}{\text{Amount of Response}} \tag{2}$$

### 3.5. Sample Collection and Handling

In order to find antibiotic residues in surface waters, the separation techniques primarily used were Liquid Chromatography, which offers sensitivity, selectivity, and reliable results, in combination with tandem mass spectrometry (LC-MS/MS) [20]. To collect water samples for examination, stainless-steel buckets were used to collect these from the surface to 50 cm below the surface. The samples were then placed in 2 L autoclaved amber glass bottles, which were rinsed in advance with ultrapure water and methanol. To avoid any bubbles, the bottles were fully filled and capped tightly. All samples were collected in triplicate. To carry out the investigations, the samples were brought to the laboratory in an ice box. The analysis was performed as per the procedure carried out by [1], with some modifications in the existing procedure.

### 3.6. Extraction of Antibiotics

A total of 500 mL of water was processed by adding 200 mL of chloroform and processing the same for 30 min at room temperature in a shaking incubator. The wastewater

was divided into two phases, where the upper extractant phase contained antibiotics and the lower phase was the separation assembly. To fully extract the antibiotics, the procedure was carried out twice. The extracted phase (lower phase), which included the antibiotics, was mixed with 50 mL of chloroform in the following two repeats, and the incubator was shaken for 30 min each time. Thus, the final amount of chloroform was 300 mL. At a temperature of 50 °C, it was dried almost completely in a rotary evaporator, and then the solid portion was mixed with 2.5 mL of methanol and injected into UPLC sampling vials, using 3 mL sterile syringes and syringe filters of 0.45 μm [41]. By using the optimized UPLC/MS technique, the samples were subsequently processed.

### 3.7. Statistical Analysis

In order to analyze the data statistically, the ANOVA test was used to determine the difference ($p < 0.05$) between the pre-monsoon and post-monsoon concentration of each antibiotic. Afterward, box plots were polluted for the targeted antibiotics. Seasonal fluctuations in antibiotic concentration were determined by the correlation coefficient, which clearly indicated a higher concentration of drugs in the pre-monsoon concentration than the post-monsoon concentration.

## 4. Results and Discussion

### 4.1. Physicochemical Characteristics of Samples

The physiochemical properties, such as pH value, total suspended particles, BOD, COD, phosphate, lead, nitrates, and alkalinity, during the pre-monsoon and post-monsoon periods of river water samples were analyzed and are listed in Table 3. All the analyses, including COD, TSS, $BOD_5$, Phosphate, Nitrate, and Alkalinity, were conducted according to the standard method for water and wastewater [42]. COD measurement was performed using the close reflex methodology and by Hach reagent, using DR-500 UV/Visible spectrophotometer, and dissolved oxygen measurement was performed continuously using an online Hach-attached probe. All the parameters were analyzed (mg/L) at the laboratory except pH (in situ). The pH is known to remain relatively neutral over this time at all the sampling points. Near-neutral pH values were found in the current study, which is comparable with studies conducted on Indian rivers. The values include the Yamuna River (6.8–7.2), River Narmada (6.5–8.9), Sabarmati River (6.5–9.3), and Ganga River (6.54–7.98) [9,43,44]. The mean concentration of TSS (total suspended solids) was found to vary from 26.56 to 47.45 mg/L, and higher during the post-monsoon period than the pre-monsoon period. TSS concentrations more than 15 mg/L in surface water were reported by the authors [45]. Pre-monsoon periods were reported to have higher BOD and COD concentrations than in the post-monsoon periods. The lower concentration during the post-monsoon period may be due to monsoonal flooding or the dilution effect. From the analyzed site, the lowest BOD (22.32) was found at Site-1 but was higher than that at reference sampling site-6 during the post-monsoon period. However, 3 mg/L is the highest permitted limit of BOD for bathing. As per the Bureau of Indian Standards (BIS), the permissible limits for surface water characteristics is: pH (6–8.5), BOD (>5 mg/L), nitrate (>50 mg/L), and phosphate (>0.1 mg/L) [9]. The highest COD (112.36) was found during the pre-monsoon period at site-5 (Okhla barrage) as most of the industrial discharge and waste from the city enters here. The release of insufficiently, or untreated, industrial effluents may be the cause of the highest phosphate concentrations discovered in Okhla barrage and Wazirabad, respectively. The main causes of pollution in surface water of the River Yamuna are nitrates and phosphates, which come from fertilizers, industrial chemicals, organic materials, and animal and human waste. Nitrate concentrations were determined to be below Bureau of Indian Standards (BIS) acceptable limits in all of the river samples [9]. By creating an algal bloom and eutrophication, the greater phosphate and nitrate concentration in the river water can reduce its quality. This is caused by chemicals being released into the river, such as pesticides, industrial chemicals, and pharmaceuticals [46]. The variability in pH values caused the alkalinity to change from one measurement point to another over the two

seasons. In general, physicochemical features have the power to either increase or decrease the intensity in terms of volumes, concentration, toxicities, and resistance to antibiotics among the micro-organisms found in water. The presence of chemicals that cause high values for physical, chemical, and biological characteristics has led to the conclusion that inadequate wastewater treatment can contribute to microbial antibiotic resistance [1].

**Table 3.** Values of physicochemical parameters (mg/L, excluding pH) of water samples taken from the Yamuna River in five different areas during the pre- and post-monsoon periods.

| Sampling Seasons | Sampling Locations | pH | TSS | BOD | COD | Phosphates | Nitrates | Alkalinity |
|---|---|---|---|---|---|---|---|---|
| **Pre-monsoon** | Site-1 | 7.03 | 36.62 | 23.68 | 98.24 | 0.91 | 3.61 | 178 |
| | Site-2 | 7.24 | 35.25 | 32.24 | 82.69 | 2.02 | 5.32 | 347 |
| | Site-3 | 7.19 | 26.56 | 38.19 | 74.36 | 2.14 | 4.85 | 354 |
| | Site-4 | 7.38 | 28.31 | 39.81 | 82.58 | 3.63 | 6.72 | 358 |
| | Site-5 | 7.29 | 38.94 | 45.75 | 112.36 | 4.57 | 7.26 | 398 |
| | Average | 7.226 | 33.136 | 35.934 | 90.046 | 2.654 | 5.552 | 327 |
| | Std.Dev | 0.130 | 5.404 | 8.371 | 15.174 | 1.443 | 1.466 | 85.633 |
| | Ref-Site (6) | 7.01 | 21.41 | 19.32 | 20.23 | 0.12 | 0.23 | 25.45 |
| **Post-monsoon** | Site-1 | 6.95 | 32.34 | 22.32 | 102.3 | 0.98 | 5.18 | 176 |
| | Site-2 | 7.11 | 36.04 | 24.75 | 98.42 | 3.01 | 5.22 | 329 |
| | Site-3 | 7.34 | 31.43 | 34.81 | 107.2 | 3.05 | 6.08 | 367 |
| | Site-4 | 7.28 | 33.41 | 41.68 | 102.4 | 2.76 | 6.45 | 356 |
| | Site-5 | 7.13 | 47.45 | 43.32 | 105.8 | 3.07 | 7.12 | 387 |
| | Average | 7.162 | 36.134 | 33.376 | 103.224 | 2.574 | 6.01 | 323 |
| | Std.Dev | 0.154 | 6.557 | 9.573 | 3.430 | 0.900 | 0.828 | 84.803 |
| | Ref-Site (6) | 6.23 | 19.2 | 17.26 | 19.31 | 0.14 | 0.19 | 23.98 |

Note: Ref-site: Jamia Millia Islamia.

*4.2. Occurrence and Concentration of Antibiotics at Different Sampling Locations*

From among the surface water samples of the River Yamuna, the concentration of antibiotics was evaluated by using a suitable technique by comparing with the selected standards. Among these selected antibiotics, the average concentration of ofloxacin in the water samples of the River Yamuna is presented in Figure 2, during pre-monsoon and post-monsoon periods. Out of the five sampling locations and one reference point in both pre-monsoon and post-monsoon periods, the ofloxacin antibiotic concentration was found at all sites except reference site locations. However, during the pre-monsoon season, the highest ofloxacin concentration of about 145.379 ng/mL was found at (Site-5) Okhla barrage and the lowest concentration was found at Wazirabad barrage of about 14.374 ng/mL. During the post-monsoon period, the maximum concentration of ofloxacin antibiotic was seen as 75.568 ng/mL at Site-5, followed by Saria Kali Khan and ITO, with the lowest concentration detected at Wazirabad barrage. Our findings are also in accordance with those of previous findings carried on the wastewater streams discharged directly into the river ecosystem with an average concentration found as about 1726.5 ng/L [9]. Additionally, the author previously reported that the highest ofloxacin level in European surface waters was five times greater (8770 ng/L) [47]. The range of ofloxacin concentrations in India, according to the author, was about 180 ng/L to 10,000 ng/L [15]. A similar investigation carried out by the author looked at the maximum occurrence of antibiotics from the fluoroquinolone group in the Musi River, in Hyderabad, India. These antibiotics include ciprofloxacin, lomefloxacin, ofloxacin, norfloxacin, enrofloxacin, and pefloxacin. Drug concentrations are increased by ineffective treatment in the industry. Of the 7 fluoroquinolones, ciprofloxacin (5015 g/L) has the highest concentration, followed by ofloxacin (542.4 g/L) and norfloxacin (251 g/L) [48]. All antibiotic concentrations in river samples are higher during the pre-monsoon period with a maximum at Okhla barrage, the last point of the River Yamuna in Delhi, and receives the highest pollution load of the entire city.

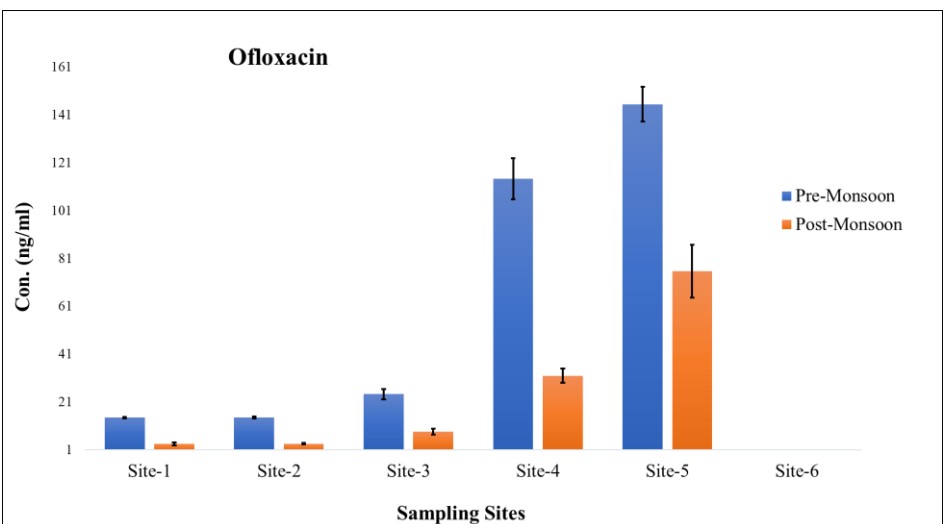

**Figure 2.** Ofloxacin concentration in surface water samples from various areas along the Yamuna River in Delhi, India.

Figure 3 displays various concentrations of antibiotic (amoxicillin) in surface water samples of the River Yamuna taken during the pre-monsoon and post-monsoon seasons of the year. Out of the total sampling sites, the antibiotic was detected at all locations with average values ranging from 3.033 ng/mL to 1.237 ng/mL. During the pre-monsoon period, the highest average concentration of amoxicillin was estimated at Okhla barrage followed by Saria Kali Khan (1.85 ng/mL), as compared to the reference sampling site, where no concentration of the given antibiotic was detected. In the post-monsoon season, the highest average concentration of amoxicillin (about 1.2347 ng/mL) was detected only at Okhla barrage followed by ISBT. Amoxicillin has been reported in a variety of concentrations all around the world. For instance, Italy's surface waters had a greater concentration of about 5700 ng/L [49,50]. Amoxicillin levels in South Africa were observed to be around 800 ng/L [51]. As seen in Figure 4, the amount of the macrolide antibiotic Erythromycin in the surface water of the River Yamuna also varied greatly. Out of the total sampling sites, and one reference site-6 (Jamia Millia Islamia), all five locations revealed the presence of antibiotics throughout the study period, with the highest average concentration at Okhla barrage (2.171 ng/mL) followed by site-3 during the pre-monsoon period. In the post-monsoon period, the highest average concentration (2.171 ng/mL) was reported at site 5, lower than those reported in China 14,180 ng/L [34,52,53] and in France (26,800 ng/L) [54].

One of the most common antibiotics that are released into the environment from both human and animal feces in their unmetabolized state are fluoroquinolones and other -lactam antibiotics. The highest concentration of ofloxacin antibiotic is due to the fact that fluoroquinolones are more stable because of their structure and functional group. This is consistent with previous findings of the rivers from Argentina, China, French, India, and the US [55]. On the other hand, researchers have discovered two key causes for this change: increased wastewater effluents and the sorption or desorption of pharmaceuticals onto or from river sediments [45]. When the influent load exceeds the treatment capacity, the WWTPs release the excess flow directly into the river without any treatment [54]. Pharmaceuticals are obviously trending in the direction of the sample locations, with concentrations rising near the Okhla barrage, primarily owing to bioaccumulation. Furthermore, the delivery of drugs from aquatic species to terrestrial organisms via the food chain may have the effects of bioaccumulation and biomagnification [28,56]. Antibiotics were found in all samples, except for those that were present in negligible levels or were beyond the instrument's range of detection. From the current investigations, the most common antibiotics found were those belonging to the fluoroquinolone group, followed by β-lactams and macrolides. Among the selected antibiotics, the detection percentage was

that of ofloxacin >amoxicillin > erythromycin. Even in soil with a high sorption capacity and high fluoroquinolone bioavailability, they accumulate and persist in the environment as a result of their poor biodegradability [1,12,14].

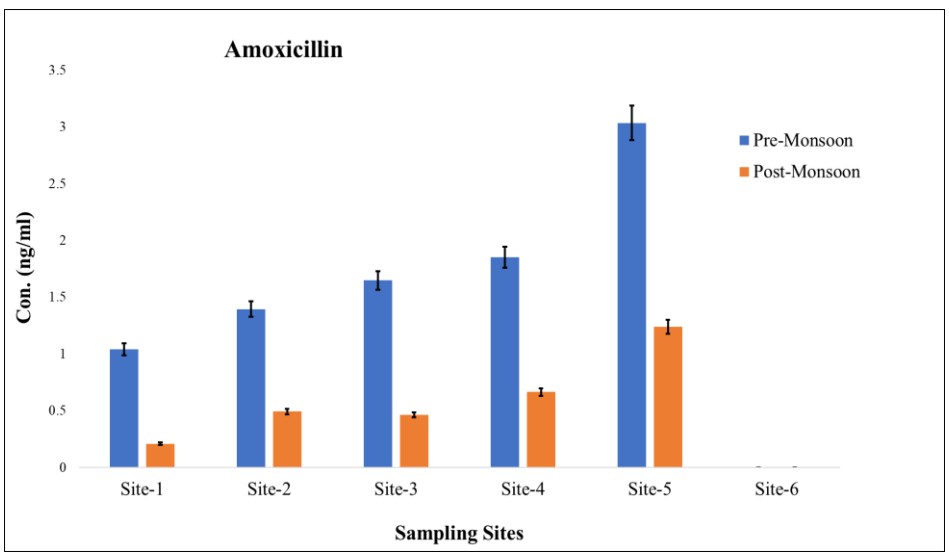

**Figure 3.** Amoxicillin concentration in surface water samples from various areas along the Yamuna River in Delhi, India.

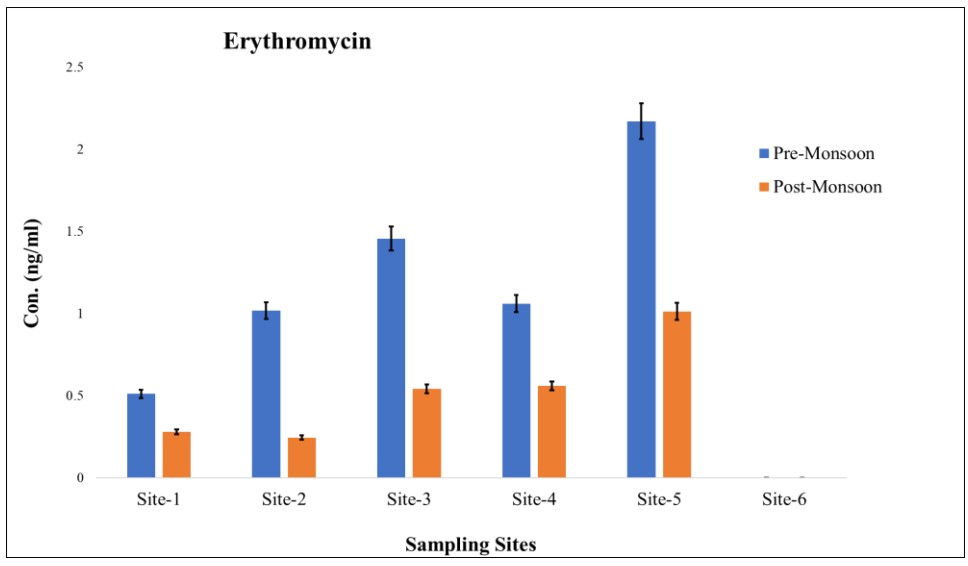

**Figure 4.** Erythromycin concentration in surface water samples from various areas along the Yamuna River in Delhi, India.

*4.3. Potential Risk of Developing Antibiotic Resistance*

Although few antibiotics were analyzed in our study during the sampling period, it is known that antibiotics are used extensively for the treatment of various diseases and can accumulate in the environment. This may be because hydrophilicity results in a lower toxicity to aquatic organisms [39]. Among the significant number of drugs that were previously evaluated [57], many of the antibiotics were included in the list of high-priority pharmaceuticals. The selected antibiotics were analyzed by using the risk quotient (RQ) approach. The risk assessment approach indicates that the drug residues significantly pose high risks to the aquatic environment. According to the risk assessment approach, the value of RQ < 0.1 does not show any ecological risk, $0.1 \leq RQ < 1$ shows

low danger, $1 \leq$ RQ < 10 exhibits moderate risk, and RQ $\geq$ 10 exhibits severe ecological danger [58,59]. The RQs estimated for three high-priority antibiotics from the predicted no-effect concentration (PNEC) in the five sample locations are shown in Figure 5, and these are denoted by various colors, depending on the level of danger these pose. The minimum concentration below which hazardous and detrimental effects have not been observed is known as the Predicted No-Effect Concentration (PNEC). The occurrence and amounts detected in river water samples taken from several areas along the Yamuna River in Delhi, India, for the current study justify the severity of the condition in the area. Out of the studied antibiotics, ofloxacin, indicated significant environmental threat (indicated by the red color in Figure 5) throughout the investigated period. High levels of antibiotics revealed an insufficient wastewater treatment system and the lack of separation of hospital, industrial, and municipal wastewater effluents before discharge into nearby surface water bodies [60]. These concentrations of drugs serve as evidence of the existence of microorganisms that are resistant to them in aquatic ecosystems. This may also be related to how some antibiotics build-up in the body. For instance, the high quantities of ofloxacin found in both the literature and the current study may be a result of the drug's persistent properties, greater stability than less biodegradation, and large accumulation in various environmental matrices. As a result, when numerous sources add antibiotics to river water pipes, its concentration rises [37]. Therefore, both low and high concentrations in environmental compartments are unacceptable in the light as they will considerably harm the ecosystem. Sensitive and useful bacteria can be killed by extremely high concentrations, leaving only the resistant microorganisms in the surface water stream. On the other hand, a low concentration of antibiotics would give the chance for microorganisms to progressively evolve resistance toward antibiotics. Eventually, the surface water stream would develop into a source of multidrug-resistant bacteria [1]. To prevent the development of antibiotic resistance in microorganisms and humans and the spread of multidrug-resistant diseases, which can be fatal, these dangers of acquiring resistance must be considered seriously. To remedy this, the necessary step might be to put in place rigorous restrictions for the handling and disposal of effluents to reduce the harmful impact of antibiotics on the living environment [9].

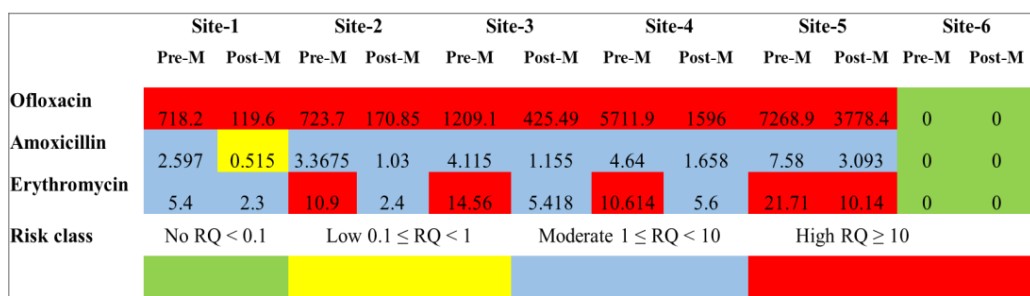

**Figure 5.** Risk Quotient of antibiotics compounds in the Yamuna River.

## 5. Conclusions and Perspectives

A reliable and universal approach to the measurement of various antibiotics in the surface of the River Yamuna was utilized. To obtain a clear image of the various concentrations of the selected groups of antibiotics in the surface water, the UPLC instrument was developed to analyze the targeted samples. The study found that the most widely used group of antibiotics are present in surface waters of urban areas. In this study, the pre- and post-monsoon presence of antibiotics showed that the main source of these chemicals was the ineffective operation of WWTPs (Wastewater Treatment Plants). The pre-monsoon concentration was higher than the post-monsoon concentration, which may be related to summer's lower dilution than that in winters. Ofloxacin was shown to have the greatest concentration of all the examined antibiotics during the pre-monsoon season. The minimum concentration of drugs (Site-1) in the upper stream of the Wazirabad area indicates

the least interaction of the heavy pollution load with the river body. From the sampling point Site-3 onward, the quality further degraded by the raw sewage from slum areas or unauthorized sewer connections brought by various drains. The fifth sampling point (Site-5) exhibits maximum pharmaceutical load as it also represents the load of 22 drains and inefficient wastewater treatment plants. The outcome of the presented study revealed that the effluent of all the major drains as well as the wastewater directly released from the urban runoff should be given a minimum tertiary treatment to meet CPCB guidelines. Drug residue levels and compositions in the aquatic ecosystem may be influenced by seasonal agricultural application, precipitation, and temperature. Uncontrolled, excessive consumption and the release of untreated wastewater into different channels may be the causes for high concentrations. It is further emphasized that some active components were within the detectable limits as these had not fully degraded. In addition, due to the lack of freshwater supplies, agricultural fields, and vegetable gardens are also irrigated with untreated surface water, which also contributes to introducing antibiotic-resistant bacteria and -resistant genes into the food chain. According to the risk assessment approach, pharmaceutical residues may represent significant threats to the aquatic ecology. The outcome of the risk assessment indicates that future research should focus on the ranking of pharmaceutical substances from a regulatory perspective. The RQ approach is superior to all other methods and is better prioritized for a particular sample location on a larger scale. Research is still needed to assess the potential dangers to the soil biota, the spread of antibiotic resistance, and crop productivity, which pose a challenge to food security and sustainability.

**Supplementary Materials:** The following supporting information can be downloaded at: https://www.mdpi.com/article/10.3390/w15030527/s1, Table S1. Selected antibiotics and their physico-chemical properties. Table S2. Studies on the existence of different antibiotics in surface waters.

**Author Contributions:** S.A. (Suriyah Akhter): Conceptualization, Data Curation, Investigations, Methodology, Writing—Original Draft Preparation, Formal Analysis, Writing—Review and Editing. M.A.B.: Conceptualization, Visualization, Writing—Review and Editing, Funding Acquisition. S.A. (Sirajuddin Ahmed): Conceptualization, Supervision, Validation, Visualization, Writing—Review and Editing. W.A.S.: Supervision, Validation, Visualization. S.A. (Sayeed Ahmad): Supervision, Validation, Visualization, H.S.: Data Curation, Investigations, Validation, Visualization. All authors have read and agreed to the published version of the manuscript.

**Funding:** This research received no external funding.

**Data Availability Statement:** All the data used and generated in this study are available from the corresponding author upon reasonable request.

**Acknowledgments:** The authors of the current work wish to acknowledge the Central Instrumentation Facility, Jamia Millia Islamia for providing facilities of LC-MS and lab support. We would like to thank all four anonymous reviewers whose constructive comments helped us to improve our manuscript and special thanks to Academic Editor for providing valuable suggestions for improving this work.

**Conflicts of Interest:** The authors affirm that they have no known financial or interpersonal conflict that would have appeared to have an impact on the research presented in this publication.

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
