# Peer review of "Profiling of Antibiotic Residues in Surface Water of River Yamuna Stretch Passing through Delhi, India"

_water, doi:10.3390/w15030527_

Round 1
Reviewer 1 Report
Dear Authors,
The authors of the manuscript "Profiling of antibiotic residues in surface water of river Yamuna stretch passing through Delhi, India" provide an overview of different aspects of antibiotic pollution in the aquatic environment. Within their article in introduction the authors address sources of antibiotic pollution, effects on environment and concentrations of antibiotic detected in the rivers.
The presented data is not really connected to each other. For example, a lot of information in the tables are shown detailing e.g. concentration of antibiotic, but in my opinion no kind of a further data-analysis was tried to e.g. correlate concentrations to population density. The data were collected and presented but no further analysis was performed . As well the authors did not really address scientific gaps, such as a need for analytical methods for the detection of low levels of antibiotics. But done research anitibiotics concetration one of the largest rivers in northern India the Yamunas River is very interesting and helpful for people working in the field of antibiotic pollutants . A study was carried out to monitor and quantify three common antibiotics in surface water, with the help of Ultra-Performance Liquid Chromatography (UPLC), coupled with Mass spectrophotometer (MS/MS) based method.
In sum, while reading the submitted manuscript, inaccuracies were noticed, which I recommend to fix (according to the comment above).
In addition below, a few editorial comments:
1. Lack of statistical analysis – error bars (figure 2, 3 i 4)
2. Check in manuscript the typos and spaces, and the size of the space between lines (especially in references).
I appreciate the manuscript presented. The topic is interested, however, some aspects must be revised before the acceptation for this Journal.
Author Response
Authors are thankful to the reviewers for their useful suggestions and comments. In the light of these comments, the manuscript is now being carefully revised. The rebuttals for the comments are given below. All queries raised by reviewers have been addressed and highlighted in the revised manuscript.
Reviewer’s Comment
Comment 1. Correlate concentrations given in the tables to population density.
Response: Authors are thankful to the reviewer for such an important suggestion. However, in this regard we would like to inform that the given tables represent the status of the drugs found in different river ecosystem of the world. It will be difficult to corelate the concentration with the population density as their geo-spatial location of sampling sites are not given in all the reference papers cited e.g., as in reference [64].
Comment 2. The authors did not really address scientific gaps, such as a need for analytical methods for the detection of low levels of antibiotics.
Response: Authors are thankful to the reviewer for an important comment. In this case, the authors want to convey that the need for the detection of low levels of antibiotics has been improved carefully and incorporated into the manuscript. The detailed comparative study incorporated is discussed below and highlighted.
“The pharmaceutical residues are ubiquitously present and have potential implications in the aquatic environment affecting the water quality and lower organisms. Since they are reportedly present in lower concentrations and thus their effect can't be seen immediately but are accumulating, which may have chronic effects in the future. Advanced analytical technologies have enabled the detection of antibiotics at extremely low concentrations (ng/L – μg/L) in different water matrices like surface waters, wastewater effluents, soil, sediments, and sludge. There is a need to identify high-priority pharmaceuticals having the potential to cause a major impact on the environment”. All this additional information has been added to the revised manuscript.
To
Allison Wu
Editor
Journal: Water
Subject: Submission of rebuttal for the manuscript Ms. I.D: Water-2059218
Dear Dr. Wu
Authors are thankful to the editor for their useful comments and suggestions. In the light of these comments, the manuscript is now being carefully revised. We have taken care of all the issues and suggestions with the manuscript pointed out by the editor The rebuttals of all the comments and suggestions are also analysed carefully. All queries raised by reviewers have been addressed and highlighted in the revised manuscript. We hope that the worthy editor will acknowledge the responses we have provided in the rebuttal letter and the revised manuscript will meet the standard of your esteemed journal “Water” and will find a place for its publication. in the revised manuscript. We earnestly thank the editor as well as reviewers for helping us to improve this manuscript.
#Editorial comments
Comment 1. Lack of statistical analysis – error bars (figure 2, 3, 4).
Response: In the light of the honorable editor’s comment, we are very thankful for such an important comment and hence in this we have revised our data to carry out statistical analysis like average, standard deviation as given in revised manuscript.
Comment 2. Check in manuscript the typos and spaces, and the size of the space between lines (especially in references).
Response: In the light of the learned editor’s comment, the author has carefully checked the typos, spaces between lines particularly in the reference section and have made the due corrections within the revised manuscript. Hope now the paper will meet the standard of the Journal for its publications.
Reviewer 2 Report
The manuscript presents the detection of emerging contaminants antibiotic-type in a surface water body using advanced chromatography and mass techniques. The manuscript is original and of interest to the audience of the journal. However, some aspects must be corrected before publication.
Comments about the work are presented below:
Introduction:
In the introduction, the subtitle of line 80, tables 1 and 2 should not be placed. The reading is interrupted for Table1 and 2. Tables 1 and 2 have important information and should go as supplementary material.
Materials and methods
In line 149, figure 1 should be improved. A scale and a legend should be placed with more specific colors of all the components of the map, sampling points and the Yumana River. The map should not have lines highlighting the area; just place a view of the location that is global, regional and of the river as such.
On line 181, enter the full name of the technique and not the initials
In the methodology, it is not clear which standard method was followed for the water quality analyzes that are presented as a result in Table 5.
Results
Figures 2, 3 and 4 need to be improved. Add error bars with standard deviation. Name the X and Y axes within the graph design. In the axis of the sampling site, standardize the nomenclature and place it in figure 1 and table 5, that is, it is Site-1, Site-2 ... Site-6.
Author Response
Authors are thankful to the reviewers for their useful suggestions and comments. In the light of these comments, the manuscript is now being carefully revised. The rebuttals for the comments are given below. All queries raised by reviewers have been addressed and highlighted in the revised manuscript.
#Reviewer’s Comment
Comment 1: In the introduction, the subtitle of line 80, tables 1 and 2 should not be placed. The reading is interrupted for Table1 and 2. Tables 1 and 2 have important information and should go as supplementary material.
Response: Thank you for your valuable suggestion, authors have moved the said tables in supplementary materials file as Table S1 and Table S2.
Comment 2. # Materials and methods: In line 192, figure 1 should be improved. A scale and a legend should be placed with more specific colors of all the components of the map, sampling points and the Yumana River. The map should not have lines highlighting the area; just place a view of the location that is global, regional and of the river as such.
Response: As per the comment of reviewer, we have carefully improved all the components of the map, scale and legends are also placed with more specific colours. The map now presents the global and regional view of the sampling locations and the river by improving the quality with the help of Arc GIS software. All this additional information has been added to the revised manuscript.
Comment 3. On line 181, enter the full name of the technique and not the initials
Response: Thank you so much for your suggestion and comments. We have modified our manuscript and made the valuable change as per the guidance of reviewer by inserting the full name of technique “Ultra-Performance Liquid Chromatography” (UPLC).
Comment 4. In the methodology, it is not clear which standard method was followed for the water quality analyzes that are presented as a result in Table 5.
Response: Thank you for such an important comment about standard methods followed for the water quality assessment. However, in this regard we would like to inform that APHA standard methods were followed for assessment of physico-chemical properties of water that are discussed and highlighted in the manuscript. All this information has been added to the revised manuscript.
Comment 5. Figures 2, 3 and 4 need to be improved. Add error bars with standard deviation. Name the X and Y axes within the graph design. In the axis of the sampling site, standardize the nomenclature and place it in figure 1 and table 5, that is, it is Site-1, Site-2 ... Site-6
Response: We thank the worthy reviewer for critically analyzing our work and making valuable comment for improvement. In this light we have carefully improved all the figures with error bars and standard deviations All the standardize nomenclature are placed within the figures as per the comment of the learned reviewers. All the changes have been incorporated into the manuscript
Reviewer 3 Report
Dear editor and authors,
Thank you for inviting me to evaluate the article (2059218) titled “Profiling of antibiotic residues in surface water of river Yamuna stretch passing through Delhi, India”. The study was carried out to monitor and quantify three common antibiotics in Yamuna river, which is one of the largest rivers in northern India. This is an interesting study. However, there are several issues that need to be explained, and supplement. My detailed comments are as follows:
1.Please explain the reason why the antibiotic residues were compared during the pre-monsoon and post-monsoon period.
2.Please add the data such as blank recovery rate and standard recovery rate of UPLC/MS/MS.
3. How about the recovery rate of extraction method used in this study?
Author Response
Authors are thankful to the reviewers for their useful suggestions and comments. In the light of these comments, the manuscript is now being carefully revised. The rebuttals for the comments are given below. All queries raised by reviewers have been addressed and highlighted in the revised manuscript.
Reviewer’s Comment
Comment 1: Please explain the reason why the antibiotic residues were compared during the pre-monsoon and post-monsoon period.
Response: Authors are thankful to the learned reviewer for the above given valuable comment. One of the reasons for comparison of antibiotic residues is to understand, what is the range of drugs during pre-monsoon and post-monsoon season of the year. Since, there is seasonal fluctuation of drug residues in the river ecosystem. During pre-monsoon season, the flow of river decreases which results the maximum concentration of drug residues in the river. During post-monsoon season, due to excessive dilution and rigorous flow caused by the nearby flooding, the concentration of drug residues decreases. Seasonal agricultural application, rainfall, and temperature could influence the levels and compositions of drug residue in the aquatic ecosystem. Risk assessments indicated that pharmaceutical residues potentially posed high risks to the aquatic ecosystem.
Comment 2: Please add the data such as blank recovery rate and standard recovery rate of UPLC-MS/MS.
Response: We thank the worthy reviewer for pointing out this issue. Here is the data of blank recovery rate and standard recovery rate of UPLC-MS/MS. In case of blank extraction rate, we compare the third concentration in linearity (L3) which is 25ppb verses post spike concentration. For the standard recovery rate, the data is given below in which L1=5 ppb, L2= 10 ppb, L3=25 ppb, L4=.50 ppb, L5=100 ppb, L6=200 ppb
1. Recovery rate of extraction method for Ofloxacin at L3 Level |
|
||||||||
Name |
Area |
||||||||
Recovery rate for method_01 |
4290 |
||||||||
Recovery rate for method_02 |
4379 |
||||||||
Recovery rate for method_03 |
4303 |
||||||||
Recovery rate for method_04 |
4286 |
||||||||
Recovery rate for method_05 |
4343 |
||||||||
Recovery rate for method_06 |
4294 |
||||||||
Recovery rate for method_07 |
4279 |
||||||||
Recovery rate for method_08 |
4262 |
||||||||
Avg Area |
4304.5 |
||||||||
Area at L3 in linearity curve |
5054 |
||||||||
% Recovery |
85.17 |
||||||||
2. Recovery rate of extraction method for Amoxicillin at L3 Level |
|
||||||||
|
|||||||||
Name |
Area |
|
|||||||
Recovery rate for method_01 |
4678 |
|
|||||||
Recovery rate for method_02 |
4720 |
|
|||||||
Recovery rate for method_03 |
4690 |
|
|||||||
Recovery rate for method_04 |
4669 |
|
|||||||
Recovery rate for method_05 |
4702 |
|
|||||||
Recovery rate for method_06 |
4759 |
|
|||||||
Recovery rate for method_07 |
4692 |
|
|||||||
Recovery rate for method_08 |
4742 |
|
|||||||
|
|||||||||
Avg Area |
4706.5 |
|
|||||||
Area at L3 in linearity curve |
5351 |
|
|||||||
% Recovery |
87.96 |
|
|||||||
3. Recovery rate of extraction method for Erythromycin at L3 Level |
|
||||||||
|
|||||||||
Name |
Area |
|
|||||||
Recovery rate for method_01 |
13463 |
|
|||||||
Recovery rate for method_02 |
13858 |
|
|||||||
Recovery rate for method_03 |
13846 |
|
|||||||
Recovery rate for method_04 |
12943 |
|
|||||||
Recovery rate for method_05 |
13882 |
|
|||||||
Recovery rate for method_06 |
13229 |
|
|||||||
Recovery rate for method_07 |
13924 |
|
|||||||
Recovery rate for method_08 |
12954 |
|
|||||||
|
|||||||||
Avg Area |
13512.4 |
|
|||||||
Area at L3 in linearity curve |
14570 |
|
|||||||
% Recovery |
92.74 |
|
|||||||
|
Comment 3: How about the recovery rate of extraction method used in this study?
Response: Authors are thankful to the reviewer for such an important comment. In this regard we would like to convey that the recovery rate for extraction method was calculated by 8 replicates at L3 v/s L3 in Linearity
Reviewer 4 Report
Method: 1. No statistical analysis stated
2. some of the methods (i.e: water sample characteristic and RQ analysis) were no described in methodology
Results and discussion: 1. results poorly presented
2. No LC-MS data stated in this manuscript
3. result poorly discussed
4. some of the data (RQ analysis) is not validate with real time data before author state his insight

Author Response
Authors are thankful to the reviewers for their useful suggestions and comments. In the light of these comments, the manuscript is now being carefully revised. The rebuttals for the comments are given below. All queries raised by reviewers have been addressed and highlighted in the revised manuscript.
#Reviewer’s Comment
Comment 1. No statistical analysis stated
Response: Authors are thankful to the reviewer for such an important comment about statistical analysis of the calculated date. In this view we have revised our data and performed the analysis of variance ANOVA for given data to check the significant difference between the mean of the calculations, and hence are incorporated in section 3.7 of manuscript.
Comment 2. Some of the methods (i.e.: water sample characteristic and RQ analysis) were no described in methodology
Response: We thank the worthy reviewer for critically analyzing our work and providing invaluable suggestions for its improvement. The standard methods highlighting the physicochemical characteristic of water samples and risk quotient have been now incorporated to the revised manuscript
Comment 3. Results poorly presented
Response: Authors are thankful to the reviewers for this valuable comment. In this regard we would like to convey your good-self that authores have revised the results and presented in proper manner in the revised manuscript. We hope that the worthy reviewer will acknowledge the given responses. We earnestly thank the reviewer for helping us to improve the results.
Comment 4. No LC-MS data stated in this manuscript
Response: Authors are thankful to the reviewers for this important comment. However, in this regard the authors want to inform that Table 2 present a data obtained from the mass transitions of LC-MS analysis.
Table 2: Mass transitions of antibiotics in river water of the Yamuna (UPLC-MS)
Antibiotics |
Calibration curve |
R2 |
Precursor ion m/z) |
Product ion (m/z) |
LOD (ng/ml) |
LOQ (ng/ml) |
Ofloxacin |
y = 269.05x +1172.69 |
0.992 |
362.3 |
318.3 |
1.5 ng/ml |
5 ng/ml |
Amoxicillin |
y = 221.43x +227.59 |
0.999 |
366.2 |
114 |
1.5 ng/ml |
5 ng/ml |
Erythromycin |
y = 644.30x +673.29 |
0.997 |
734.5 |
158.1 |
1.5 ng/ml |
5 ng/ml |
Comment 5. Result poorly discussed
Response: Authors are thankful to the reviewers for this important comment In this regard we would like to convey your good self that, in discussion part, the relevant results are properly discussed and presented appropriately in the revised manuscript.
Comment 6. Some of the data (RQ analysis) is not validate with real time data before author state his insight
Response: Authors are thankful to the learned reviewer for the above given valuable comment. However, in this regard we would like to inform that, non-contaminated area (reference site) has an RQ < 0.1 that is almost negligible. Similarly, before the entry of river Yumana in Delhi, the RQ is less than 0.1 except ofloxacin during pre-monsoon season. The value of risk quotient reduces significantly during post monsoon season of the year. We hope that the worthy reviewer will acknowledge the response we have provided in the revised manuscript.